# STATESTREAM: A TOOLBOX TO EXPLORE LAYERWISE-PARALLEL DEEP NEURAL NETWORKS

## ABSTRACT

Building deep neural networks to control autonomous agents which have to interact in real-time with the physical world, such as robots or automotive vehicles, requires a seamless integration of time into a network's architecture. The central question of this work is, how the temporal nature of reality should be reflected in the execution of a deep neural network and its components. Most artificial deep neural networks are partitioned into a directed graph of connected modules or layers and the layers themselves consist of elemental building blocks, such as single units. For most deep neural networks, all units of a layer are processed synchronously and in parallel, but layers themselves are processed in a sequential manner. In contrast, all elements of a biological neural network are processed in parallel. In this paper, we define a class of networks between these two extreme cases. These networks are executed in a streaming or synchronous layerwise-parallel manner, unlocking the layers of such networks for parallel processing. Compared to the standard layerwise-sequential deep networks, these new layerwise-parallel networks show a fundamentally different temporal behavior and flow of information, especially for networks with skip or recurrent connections. We argue that layerwise-parallel deep networks are better suited for future challenges of deep neural network design, such as large functional modularized and/or recurrent architectures as well as networks allocating different network capacities dependent on current stimulus and/or task complexity. We layout basic properties and discuss major challenges for layerwise-parallel networks. Additionally, we provide a toolbox to design, train, evaluate, and online-interact with layerwise-parallel networks.

## 1 INTRODUCTION

Over the last years, the combination of newly available large datasets, parallel computing power, and new techniques to design, implement, and train deep neural networks has led to significant improvements and numerous newly enabled applications in various fields including vision, speech, and reinforcement learning. Considering applications for which a neural network controls a system that interacts in real-time with the physical world, ranging from robots and autonomous vehicles to chat-bots and networks playing computer games, renders it essential to integrate time into the network's design.

In recent deep learning literature, enabling networks to learn and represent temporal features has gained interest. Methods were presented leveraging short-term dynamic features to build temporal consistent network responses (e.g. Ilg et al. (2017), Luc et al. (2017)) as well as networks learning to store and utilize information over longer time periods (e.g. Neil et al. (2016), Graves et al. (2016)).

Two major aspects considering the role of time in neural networks can be distinguished: First, the way neural networks and their components such as layers or single units, are implemented. For example, network components could operate sequentially or in parallel, and in case of parallel evaluation, synchronous and asynchronous implementations can be distinguished. Second, the extent to which the network through its architecture can form representations of temporal features. For example, if the network has no mechanisms to integrate information over time, such as recurrent connections, the network will not be able to represent temporal features, such as optic-flow. In this

work, we focus on the implementation aspect but highly emphasise that our approach fundamentally influences the network's temporal behavior and the way information is integrated over time.

Whereas, biological neural networks and some realizations of neural networks in silicon (reviewed in Indiveri et al. (2011), comparison in Farabet et al. (2012)) can operate on a continuous temporal dimension, we will assume a discrete (frame-based) temporal domain throughout this paper.

## 1.1 LAYERWISE-PARALLEL DEEP NEURAL NETWORKS

Considering sequential and parallel realizations of artificial neural networks, at one end of the spectrum, biologically inspired models of spiking neural networks have been studied for a long time (e.g. Liu et al. (2015)) and, in most cases, are simulated in a way that states of all neurons in a network are updated in parallel and in a synchronous frame-based manner. In contrast to this parallel processing of all neurons, modern deep neural networks are constructed using collections of neurons, sometimes called layers, modules, or nodes, and while all neurons of the same layer are computed in parallel, the layers themselves are computed sequentially.

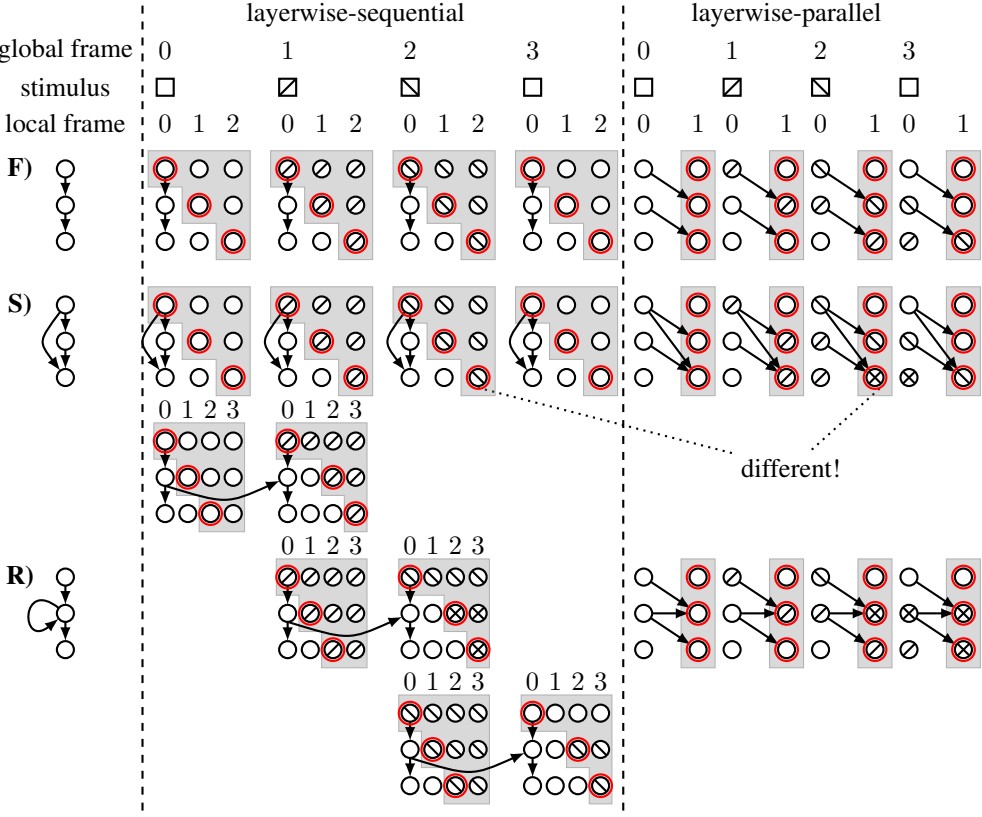

Figure 1: Three simple network examples, a pure feed forward F), a skip S), and a recurrent network R), are shown (left most column), illustrating the difference between sequential (middle column) and layerwise-parallel (right column) network execution. For both network types, inference on four succeeding (from left to right) time frames (pictograms: *empty* - / - \ - *empty*) is drawn. Encircled nodes indicate currently updated / computed / inferred layers and grey underlayed areas indicate already computed network parts. Pictograms (*empty*, /, \) inside layers indicate information from this stimulus in a layer at a specific time frame. To increase clarity for the layerwise-parallel case, we omitted information from previous stimuli still memorized by the network. For the layerwise-sequential recurrent network (bottom left network), we used a 1-step rollout window. Local frames for layerwise-sequential networks differ between architectures (3 frames for F) and S), and 4 frames for R).

In this work, we argue for a network type between these two ends of the spectrum which we call *layerwise-parallel* deep neural networks. The difference to the widely used layerwise-sequential networks in deep learning is that layers have a memory of their previous state and compute their next state on the basis of the (temporally) previous state of all (topologically) previous layers. A schematic illustration of layerwise-sequential and layerwise-parallel networks is shown in Fig. 1. We will call the part of a layer holding its current state *neuron-pool* (nodes in Fig. 1), parts of the network holding information about transformations between neuron-pools *synapse-pools* (edges in Fig. 1), and functions yielding updates for some network parameters also *plasticities* (not shown in Fig. 1). In our definition, all responses of synapse-pools targeting the same neuron-pool are simply added up. More thorough definitions are given in Section 2.

In Fig. 1, network inference is illustrated for the two network types over four succeeding time frames. In this work, *time frame* refers always to the global time frame in which stimuli are presented to the network. For layerwise-sequential networks, we have another implicit type of local frames due to their sequential nature, and the number of local frames depends on network architecture (compare Fig. 1: 3 local frames for F) and S), 4 local frames for R)). In contrast, for layerwise-parallel networks all layers are updated in parallel, leading to two local frames (current and updated). Information contained in the stimuli (squares) and neuron-pool states (circles) is encoded as *empty*, /, or \. Simple forward architectures (example in first row) without skip or recurrent connections lead to similar temporal behavior for the two network types. Introducing skip or recurrent connections (S) and R) example in Fig. 1) leads to potentially different temporal behavior for layerwise-sequential and layerwise-parallel networks. For example in Fig. 1, network responses differ between layerwise-sequential and layerwise-parallel networks at the 2. frame for the S) and R) networks (see *different!* in Fig. 1). This difference becomes more drastic considering larger, more complex network architectures, for which information from different time frames is integrated. Considering a layerwise-parallel network at a certain point in time, information from different previous time frames is distributed across the network. The distribution pattern is directly defined by the network's architecture. The biological counterpart as well as recent deep learning literature suggest to use gating mechanisms to guide content and time dependent information flow.

One aspect of layerwise-parallel networks is the synchronization of the parallel updated layers. This is important especially considering neuron-pools representing temporal features, because these, by definition, depend on temporal differences. In case of asynchronous parallelization of network parts, one solution would be to provide time itself as a network input. We focused on the synchronized approach, because otherwise networks would have to learn to use additionally provided input of time and also temporal features would be harder to interpret.

Another property of layerwise-parallel networks is, that networks are parallelized independently of their architecture. A layerwise-parallel network is designed using constrained shallow elements and the network is parallelized across all elements. For example, we prohibit using an arbitrary deep network as a synapse-pool connecting two neuron-pools. A detailed definition of neuron and synapse-pool operations is given in Section 2. With respect to this architecture-independent parallelization, layerwise-parallel networks also differ from other model-parallel approaches like synthetic gradients (Jaderberg et al., 2017) or direct feedback alignment (Nøkland, 2016).

## 1.2 RELATION TO STATE-OF-THE-ART

One major advantage of layerwise-parallel over layerwise-sequential networks is that network elements such as neuron-pools and plasticities can be computed in parallel. As stated in Jaderberg et al. (2017), the sequential nature of current deep networks results in computation locks, hindering efficient execution. Several approaches were proposed to circumvent the backward lock, which locks parameter updates due to the training procedure, providing auxiliary error signals for intermediate layers (e.g. Lillicrap et al. (2016), Jaderberg et al. (2017), Nøkland (2016)). While these methods, to some extent, solve some drawbacks of the most widely used and effective technique for neural network training, namely backpropagation, they do not address the more fundamental difference between parallel and sequential network evaluation between biological and artificial neural networks: the network's integration of temporal information during inference stays the same as for layerwise-sequential networks. Further, these approaches are not directly applicable for layerwise-parallel networks and would have to take the temporal displacement between layers into account.

Integration of temporal information in deep networks often is achieved using recurrent neural networks (RNN). For inference and training, these RNNs are normally converted into feed forward networks using network rollout and weight sharing over time (e.g. Hochreiter & Schmidhuber (1997)), especially to use existing methods to train feed forward networks. Similar to previously mentioned methods which target problems of backpropagation, also the idea of network rollout only tackles a symptom arising from layerwise-sequential networks, while creating new challenges like network initialization at beginning of rollout or scalability over time, which is not the case for our approach, at least during inference.

Beside recurrent connections, also skip connections are widely used in deep networks, such as ResNets (He et al., 2016). Especially used identity skip connections can be interpreted as a local network rollout acting as a local filtering which could also be achieved through recurrent self connections (Greff et al., 2017). Hence it seems, currently used skip connections are primarily used to mitigate problems of backpropagation rather than to form early, temporally shallow representations in abstract layers on the basis of layers with lower abstraction, which would be biologically plausible (Bullier, 2001).

The concept of layerwise-parallel networks is also strongly related to ideas like BranchyNet (Teerapittayanon et al., 2016), which use intermediate features for early classification and hence enable stimulus complexity dependent allocation of network resources. This is natively achieved with layerwise-parallel networks using skip connections, which, for layerwise-parallel networks, introduce temporal shortcuts. Hence in general, the network has shorter response times, using short (in time and network architecture) pathways, for simple, and longer response times, using deeper pathways, for complex stimuli. A simple example for this is given in Section 3.

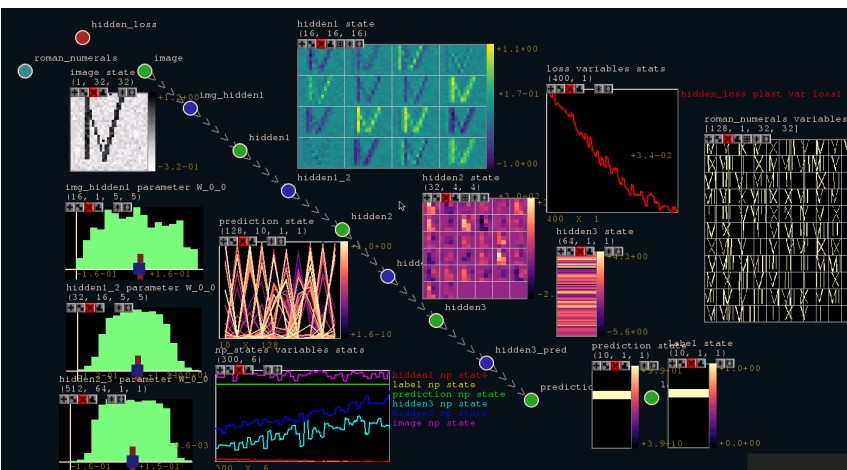

Figure 2: Visualization example of a simple classification network using the provided toolbox (best viewed in color). The network is shown as graph (green nodes are neuron-pools, blue nodes are synapse-pools) together with information about the network.

The already mentioned different integration of information over time between layerwise-sequential and layerwise-parallel networks causes also differences in network evaluation. For layerwise-parallel networks, the response on a given stimulus is delayed in time dependent on the network architecture. Hence on the one hand, existing performance measures for layerwise-sequential networks, for example, accuracies or confusion-matrices, must be adapted to take time into account, while on the other hand, measures not available for layerwise-sequential networks, such as reaction times, can now be employed.

Further, network training has to be adapted for layerwise-parallel networks. For layerwise-sequential networks, normally an error is formulated for network outputs and back-propagated through the network providing a signal to update parameters (Rumelhart et al. (1986)). To some extent, existing training mechanisms for layerwise-sequential networks can be applied to layerwise-parallel networks by using local losses, and rolling out local parts of the network. Also biologically inspired

training rules, such as Hebbian learning (Hebb, 1949), can be used. We provide more details on solutions for evaluation and training of layerwise-parallel networks in sections 3.1 and 3.2.

In the previous paragraphs, we laid out some motivation for layerwise-parallel networks and compared it to layerwise-sequential networks. And while we are not the first ones pointing out certain drawbacks of layerwise-sequential networks (e.g. Jaderberg et al. (2017), Farabet et al. (2012)), it is our understanding, that the underrepresentation of layerwise-parallel networks in deep learning literature is due to a lack of tools to explore this class of networks.

One of the main contributions of this work, is to provide an open source toolbox (available at *http://bit.ly/2yNfroI*) to design, train, evaluate, and interact with layerwise-parallel deep networks. An example screenshot of the provided graphical user interface is shown in Fig. 2.

## 2 LAYERWISE-PARALLEL DEEP NETWORKS

In this section, we give a formal definition of layerwise-parallel networks and their elements.

### 2.1 NETWORK MODEL AND INFERENCE

We describe a neural network as a graph $(\mathcal{V}, \mathcal{E})$, with $\mathcal{V} = \{v_i | i = 1, ..., N_\mathcal{V}\}$ being a set of vertices and $\mathcal{E} = \{e_j | j = 1, ..., N_\mathcal{E}\}$ being a set of directed (hyper-) edges with potentially multiple source vertices $\mathrm{src}_j \subset \mathcal{V}$ and a single target vertex $\mathrm{tgt}_j \in \mathcal{V}$. Each vertex $v_i$ has a fixed dimensionality $D_i = (F_i, W_i, H_i) \in \mathbb{N}^3$, a state $x_i^t \in \mathbb{R}^{D_i} = \mathbb{R}^{F_i} \times \mathbb{R}^{W_i} \times \mathbb{R}^{H_i}$ at time $t \in \mathbb{N}$, and a parameterized mapping $\sigma_{\vartheta_i^t}^i : \mathbb{R}^{D_i} \to \mathbb{R}^{D_i}$ with some parameters $\vartheta_i^t$. Each edge $e_j$ has a parameterized mapping

$$f_{\theta_j^t}^j : \left( \prod_{v_i \in \mathrm{src}_j} \mathbb{R}^{D_i} \right) \to \mathbb{R}^{D_{\mathrm{tgt}_j}}$$

with some parameters $\theta_j^t$. For a vertex $v_i$, let $\mathrm{input}_i \subset \mathcal{E}$ denote the set of all edges targeting $v_i$. We define a one-step temporal propagation for every vertex:

$$x_i^{t+1} = \mathcal{F}_i(x^t, \theta^t, \vartheta^t) = \sigma_{\vartheta_i^t}^i \left( \sum_{e_j \in \mathrm{input}_i} f_{\theta_j^t}^j(\{x_k^t\}_{k \in \mathrm{src}_j}) \right)$$

As stated earlier, we also refer to the vertices of the network graph as *neuron-pools* and to the edges as *synapse-pools*.

Using this one-step temporal propagation, all network states $x_i$ can be updated independently of each other and in parallel from time step to time step.

Although, the above definition is general, the provided toolbox introduces some restrictions on what update functions can be specified. An explicit specification of the internal structure of the update functions was chosen to include common elements of deep neural networks, such as convolutional layers, inception mechanisms and gated connections. Please see Appendix 6.1 for more details. We emphasize again that for layerwise-parallel networks, parallelization is independent from the network architecture. The network is designed using certain elements and these are always parallelized. Hence, these elements have to be flexible to a certain extent, to enable users to design various architectures.

### 2.2 NETWORK TRAINING

Let $\Omega^t = (\theta^t, \vartheta^t)$ denote all current parameters. We refer to a mapping $p$ that, on the basis of current and previous states and parameters, produces a parameter update for some parameters $\omega_p \subset \Omega$ as a *plasticity*:

$$\Delta\omega_p(t) = p(x^{\tau \le t}, \Omega^{\tau \le t})$$

For a given set of plasticities $\mathcal{P} = \{p_i | i = 1, ..., N_{\mathcal{P}}\}$, we define a one-step parameter update function for every single parameter $w \in \Omega$:

$$w^{t+1} = w^t + \sum_{i \text{ with } w \in \omega_{p_i}} \Delta\omega_{p_i}^w$$

Note that all plasticities can be computed independently from each other only on the basis of previous states of neuron-pools and synapse-pools, and hence in parallel to the update functions for network inference.

Beside this abstract definition, some more explicit examples of plasticities, which are provided with the toolbox, are given below.

## 3 CHALLENGES

Most challenges working with layerwise-parallel networks are caused by the fact that at a given point in time, information from different previous time frames is distributed across the network. The distribution pattern is directly given by the network's architecture and can be conveniently visualized using a rolled out version of the network. In general, gating neuron-pools could guide, for example dependent on changes in input stimuli, the information flow through this pattern.

A small example layerwise-parallel network for MNIST classification is illustrated in Fig. 3, showing the network architecture in 3a and the rolled out network in Fig. 4 to visualize information flow. Similar to the idea of BranchyNet (Teerapittayanon et al., 2016), the network uses two paths of different depths for classification. We use this small network to illustrate some important mechanisms of layerwise-parallel networks.

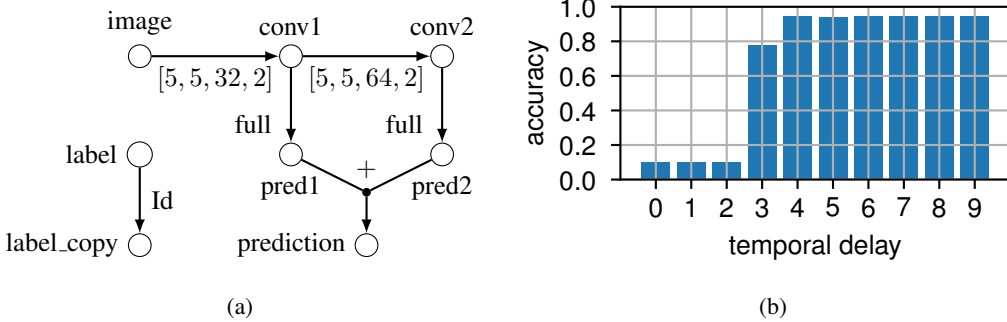

(a)                                    (b)

Figure 3: Illustration of a 2-path classification network. (a) The graph of the network. The network consists of two stacked convolutions, for which neuron-pools conv1 and conv2 ([rf, rf, features out, stride]) hold the resulting feature maps, and fully connected classification on top of both of them, with pred1 and pred2 storing the intermediate classification results. These intermediate results are aggregated through summation in the final result (prediction). For training, a one-step copy of the current ground-truth, which is stored in the neuron-pool label and label_copy, is needed. (b) Classification accuracies on MNIST test dataset, relative to temporal offset in frames between stimulus (image and label) onset and network response (prediction).

### 3.1 PLASTICITIES

As stated above, plasticities operate in parallel to neuron-pools and while all neuron-pools are computing the one-step update $t \to t+1$, all plasticities compute current parameter updates on the basis of the current time step $t$. For most plasticities, the network is rolled out locally, considering only a subset of all neuron-pools, and initialized with neuron-pool states at time $t$. This is illustrated in Fig. 4, where local rollouts are shown for the two used plasticities to train the example network in Fig. 3a. After plasticities have computed parameter updates, these updates are aggregated by the synapse-pools, and in the next step the plasticity operates on the states of the now updated neuron-pool states and parameters from time $t+1$.

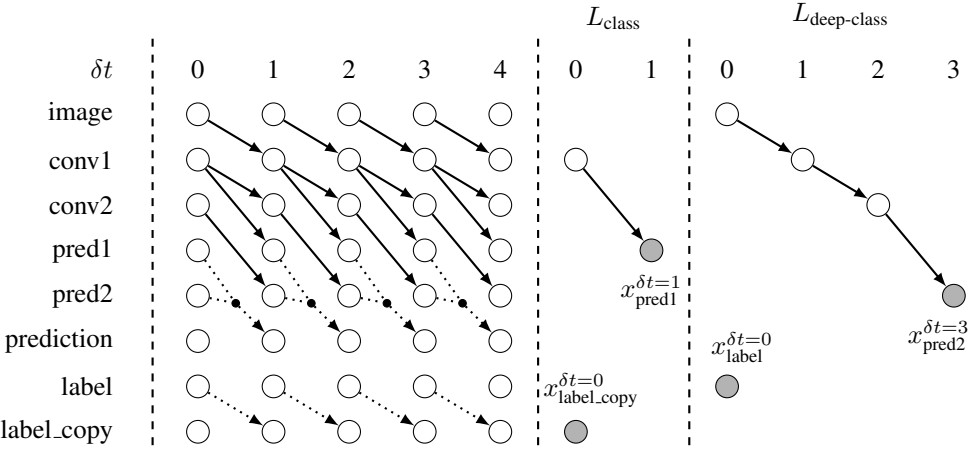

Figure 4: Rolled out network for maximal path length 4 and the two sub-networks used to compute the loss for the plasticities. All continuous lines represent synapse-pools which are initialized randomly and trained by the plasticities. All dotted lines are initialized as identity and are not trained.

To increase transparency and re-usability, we separate plasticities into two parts: An *update part*, which, state and memory free, computes a temporally local estimate of parameter updates and an *optimizer part* which transforms the update estimates, for example, through temporal smoothing, into a current update which is used for the actual update of parameters.

The toolbox provides some of the most widely used optimizers, such as stochastic gradient decent and ADAM (Kingma & Ba, 2015) and can easily be extended with new ones. Additionally, three types of update estimators are provided to specify plasticities:

**Loss based update:** To leverage the large amount of existing techniques for training of layerwise-sequential deep networks, a loss can be defined on the basis of one or two neuron-pool states at a certain temporal offset $\delta t$ from the current time step $\delta t = 0$. For example, considering the network from Fig. 4, two loss-based plasticities are used to train the network:

$$L_{\text{class}}\left(x_{\text{pred1}}^{\delta t=1}, x_{\text{label\_copy}}^{\delta t=0}\right) = \text{categorical-crossentropy}\left(x_{\text{pred1}}^{\delta t=1}, x_{\text{label\_copy}}^{\delta t=0}\right)$$

$$L_{\text{deep-class}}\left(x_{\text{pred2}}^{\delta t=3}, x_{\text{label}}^{\delta t=0}\right) = \text{categorical-crossentropy}\left(x_{\text{pred2}}^{\delta t=3}, x_{\text{label}}^{\delta t=0}\right)$$

Here, all losses are based on two neuron-pools. To compute the loss, the network is rolled back locally from the neuron-pools until *now* ($\delta t = 0$), being transformed into a feed forward network, as can be seen on the right side of Fig. 4. Training is done as usual for layerwise-sequential networks.

Note, that the validity and temporal properties of what we train, highly depend on the chosen neuron-pools and temporal offsets. Concerning validity, for example for $L_{\text{deep-class}}$ we could not have chosen neuron-pool states $x_{\text{pred2}}^{\delta t=4}$ and $x_{\text{label}}^{\delta t=1}$ because then the rolled back network would have needed an input image and label from the future time step $\delta t = 1$ which are not available now $\delta t = 0$. Concerning temporal properties, for example, if we would define the loss $L_{\text{class}}$ on $x_{\text{pred1}}^{\delta t=1}$ and $x_{\text{label}}^{\delta t=0}$ we would have introduced a temporal offset of 1 between prediction and ground-truth, leading to unintended behavior especially when the input changes.

In general, loss based plasticities are expensive in the sense that potentially, for large parts of the network, the inference (forward) step is done twice, once in the neuron-pools and potentially more than once due to the rollout in the plasticity. Hence, local plasticities are preferred, which only operate on a small set of neuron-pools and do not need a deep rollout. To achieve this, we suggest functional modularisation of the network which also increase network transparency and trainability.

**Hebbian based update:** With some restrictions, we also provide the biologically more plausible well known Hebbian learning rule (Hebb, 1949) and some of its variants as an update estimator. For example, for a synapse $w_{ij}$, connecting some neuron $x_i$ with some neuron $x_j$:

$$\Delta w_{ij}^t = x_i^t x_j^{t+1}$$

Note, that this is used as an estimate and could be used as input for some optimizer, such as ADAM. Plasticities based on this estimator always use an internal one step rollout of the target neuron-pool and hence provide rather local plasticities compared to loss based plasticities.

**Parameter regularization update:** Parameter update estimators can also directly be based on parameters rather than states, which is the case for example using $L_1$ or $L_2$ regularization on certain parameters.

### 3.2 NETWORK EVALUATION

Considering performance measures for layerwise-parallel networks, we follow general ideas from the analysis of spiking neural networks and experimental psychology (e.g. Diehl et al. (2015), Woods et al. (2015)).

Let $m(x, y)$ denote any performance measure for a layerwise-sequential deep network, for example a confusion-matrix or an accuracy value, where $y$ is the network's response for a stimulus $x$. This can be converted into a performance measure for layerwise-parallel networks:

$$m^{\delta t}(x_{t_{\text{onset}}}, y_{t_{\text{onset}}+\delta t}) = m(x_{t_{\text{onset}}}, y_{t_{\text{onset}}+\delta t})$$

Where $t_{\text{onset}}$ denotes the time of a stimulus onset (first frame a stimulus is presented). We measure current performance dependent on the temporal offset $\delta t$ between the network's current response and previous stimuli. On the basis of this, a concept of reaction- or response times can be defined, e.g. measuring the mean offset after which a certain performance measure reaches a given threshold. An example of a time dependent accuracy evaluation for the 2-path network from Fig. 3a is given in Fig. 3b. Due to the network's architecture, performance is at chance level for the first three time steps. Then information about the stimulus reaches the prediction neuron-pool through the short path before, after one additional time step, also the longer path becomes active, from which on the network reaches its highest accuracy. Stimuli were always presented for 12 consecutive frames.

## 4 THE STATESTREAM TOOLBOX

To explore layerwise-parallel deep networks, we provide an open source toolbox enabling design, training, evaluation, and interaction with this kind of networks. Networks are specified in a text file, and a core process distributes the network elements onto separate processes and/or GPUs. Elements are executed with alternating read and write phases, synchronized via a core process, and operate on a shared representation of the network. The toolbox is written in Python and uses the Theano (Theano Development Team, 2016) backend. The shared representation enables parallelization of operations across multiple processes and GPUs on one machine and enables online interaction.

An additional motivation for intuitive, direct, and adjustable interaction with networks is that current deep learning literature (e.g. Vertens et al. (2017), Gupta et al. (2017), Marblestone et al. (2016)) suggests that network architectures will become more complex and heterogeneous. These functional modularized architectures increase network understanding through transparent auxiliary neural interfaces, such as occupancy grids or optic flows, and trainability, using local losses to train sub-networks. Understanding of these network's internal dynamics is important, concerning debugging and optimizing architectures as well as safety aspects and to guide the design of future architectures.

The chosen implementation of layerwise-parallel networks favors certain network architectures. For example, the overall frame rate of the network primarily depends on the slowest network element (neuron-pool or plasticity) rather than on the overall number of elements, as long as sufficient computation resources are available. With this toolbox, we did not intent to compete with existing deep learning frameworks with respect to memory consumption or training speed but rather provide the software infrastructure to explore layerwise-parallel deep networks, which, to our knowledge, other deep learning software does not.

## 5 CONCLUSION

In this paper, we defined and discussed layerwise-parallel deep neural networks, by which layerwise model-parallelism is realized for deep networks independently of their architecture. We argued that layerwise-parallel networks are beneficial for future trends in deep network design, such as large functional modularized or recurrent architectures as well as for networks allocating different network capacities dependent on stimulus and/or task complexity. Due to their biologically inspired increased parallelizability, layerwise-parallel networks can be distributed across several processes or GPUs natively without the need to explicitly specifying the network parts which should be parallelized. Finally, we presented an open source toolbox to explore layerwise-parallel networks providing design, training, evaluation, and interaction mechanisms.

We would like to think of this work as a step towards native model-parallel deep networks, connecting the networks architecture directly to the temporal domain. For this, major challenges for the future remain, such as a more general formulation of neuron and synapse-pools than the one used in the provided toolbox, the design of new local plasticities, or designing more adequate tasks which take the temporal domain into account.

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

# 6 APPENDIX

## 6.1 COMPUTATION DIAGRAMS FOR NEURON AND SYNAPSE-POOLS

This section provides a detailed description of the computations performed in neuron and synapse-pools. Dependent on their specification, for most pools, large parts of this computation will not be executed.

In general, capital letters will denote some fixed dimensionality, and we will use the same capital letter to denote this dimensionality as well as the set of possible indeeces $N = \{1, ..., N\}$.

Let $\text{BS}, N, S, P \in \mathbb{N}$ denote the number of samples in a batch, neuron-pools, synapse-pools, and plasticities. For a neuron-pool $n \in N$, we notate:

- Features, Width and Height: $C_n, W_n, H_n \in \mathbb{N}$.
- Layer activation: $y_n(t) \in Y_n := \mathbb{R}^{\text{BS} \times C_n \times W_n \times H_n}$ at frame $t \in \mathbb{N}$.
- Activation function: $\sigma_n^{\text{np}} : Y_n \to Y_n$.
- Noise function: $\varepsilon_n^{\text{np}} : Y_n \to Y_n$.
- Bias $b_n^t \in \mathbb{R}^{C_n}$.
- Gain $g_n^t \in \mathbb{R}^{C_n}$.

For synapse-pool $s \in S$, we denote:

- Number of input factors: $F_s \in \mathbb{N}$.
- Number of summands for factor $f \in F_s$: $A_{s,f} \in \mathbb{N}$.
- Target neuron-pool: $n_s^{\text{TGT}} \in N$
- Source neuron-pools: $n_{s,f,a}^{\text{SRC}} \in N$, for $f \in F_s$ and $a \in A_{s,f}$
- Activation function for factor $f \in F_s$: $\sigma_{s,f}^{\text{sp}} : Y_{n_s^{\text{TGT}}} \to Y_{n_s^{\text{TGT}}}$.
- Synaptic weights: $w_{s,f,a} \in \mathbb{R}^{C_{n_s^{\text{TGT}}} \times C_{n_{s,f,a}^{\text{SRC}}} \times \text{rf} \times \text{rf}}$, for $f \in F_s$ and $a \in A_{s,f}$
- Pre-processing weights: $p_{s,f,a} \in \mathbb{R}^{C_{n_s^{\text{TGT}}} \times C_{n_{s,f,a}^{\text{SRC}}} \times 1 \times 1}$, for $f \in F_s$ and $a \in A_{s,f}$
- Pool-conv-upsample pre-process function: $\text{pcu-ppp}(.|w, p) : Y_{n_{s,f,a}^{\text{SRC}}} \to Y_{n_s^{\text{TGT}}}$, with:

$$\text{pcu-ppp}(y\|w, p) = \delta_{\text{pool}} \left( w \star \delta_{\text{upsample}}(p \star y) \right)$$

  where $\star$ means convolution, $\delta_{\text{pool}}$ means downsampling if target space is smaller than source space using adequate strides, and $\delta_{\text{upsample}}$ means upsampling if target space is larger than source space repeating activations in space.

As mentioned, all network elements operate synchronously alternating between a read and a write phase. Considering only network inference, the following pseudocode describes the read operations for all neuron-pools. Note, that the outer loop over neuron-pools parallelizes.

**for** $n \in N$ **do**
    *# Read NP parameter: bias and gain.*
    $\widetilde{b}_n \leftarrow b_n^t$
    $\widetilde{g}_n \leftarrow g_n^t$
    *# Loop over all sources.*
    **for** $s \in S$ **do**
        **if** $n_s^{\text{TGT}} = n$ **then**
            *# Read input NP states and SP parameter.*
            **for** $f \in F_s$ and $a \in A_{s,f}$ **do**
                $\widetilde{y}_{s,f,a} \leftarrow y_{n_{s,f,a}^{\text{SRC}}}^t$
                $\widetilde{w}_{s,f,a} \leftarrow w_{s,f,a}^t$
                $\widetilde{p}_{s,f,a} \leftarrow p_{s,f,a}^t$
            **end for**

         **end if**
      **end for**
  **end for**

The execution / writing phase at frame $t + 1$ for all neuron-pools:

1: **for** $n \in N$ **do**
2:     *# Compute post-synaptics for input sps.*
3:     **for** $s \in S$ **do**
4:       **if** $n_s^{\text{TGT}} = n$ **then**

5: $$\widetilde{x}_s \leftarrow \prod_{f \in F_s} \sigma_{s,f}^{\text{sp}} \left( \sum_{a \in A_{s,f}} \text{pcu-ppp}\left(\widetilde{y}_{s,f,a} \| \widetilde{w}_{s,f,a}, \widetilde{p}_{s,f,a}\right) \right)$$

6:       **end if**
7:     **end for**
8:     *# Accumulate post-synaptics.*
9:     $\widetilde{y}_n \leftarrow 0$
10:    **for** $s \in S$ **do**
11:      **if** $n_s^{\text{TGT}} = n$ **then**
12:        $\widetilde{y}_n \leftarrow \widetilde{y}_n + \widetilde{x}_s$
13:      **end if**
14:    **end for**
15:    *# Normalization and gain.*
16:    $\widetilde{y}_n \leftarrow g_n(\widetilde{y}_n - m_n)/s_n$
17:    *# Noise, activation, bias.*
18:    $\widetilde{y}_n \leftarrow \sigma_n(\varepsilon_n(\widetilde{y}_n)) + b_n$
19:    *# Write to shared memory.*
20:    $y_n^{t+1} \leftarrow \widetilde{y}_n$
21: **end for**

Additionally, neuron-pools and synapse-pools also aggregate and apply updates for their parameters from plasticities.

## 6.2 EXAMPLE OF A YAML NETWORK SPECIFICATION

Here we show the specification file for the layerwise-parallel example in Fig. 3a:

```
# Demonstration example

name: demonstration_example
agents: 128
globals:
    glob_input_size: 28
neuron_pools:
    image:
        shape: [1, glob_input_size, glob_input_size]
    label:
        shape: [10, 1, 1]
    prediction:
        tags: [prediction]
    conv1:
        shape: [32, glob_input_size // 2, glob_input_size // 2]
        tags: [hidden]
    conv2:
        shape: [64, glob_input_size // 4, glob_input_size // 4]
        tags: [hidden]
    pred1:
        tags: [prediction]
    pred2:
        tags: [prediction]
```

```
    label_copy:
        shape: [10, 1, 1]
synapse_pools:
    img_c1:
        source: [[image]]
        target: conv1
        rf: [[5]]
    c1_c2:
        source: [[conv1]]
        target: conv2
        rf: [[5]]
    c1_pred:
        source: [[conv1]]
        target: pred1
    c2_pred:
        source: [[conv2]]
        target: pred2
    pred_pred:
        source: [[pred1, pred2]]
        target: prediction
        rf: [[1, 1]]
        init W_0_0: id
        init W_0_1: id
    label_cp:
        source: [[label]]
        target: label_copy
        rf: [[1]]
        init W_0_0: id
plasticities:
    class:
        type: loss
        loss_function: categorical_crossentropy
        source: pred1
        source_t: 1
        target: label_copy
        target_t: 0
        lr: 1e-3
        tags: [adam_optimizer]
        parameter:
        - [sp, c1_pred, W_0_0]
    deep_class:
        type: loss
        loss_function: categorical_crossentropy
        source: pred2
        source_t: 3
        target: label
        target_t: 0
        lr: 1e-4
        tags: [adam_optimizer]
        parameter:
        - [sp, img_c1, W_0_0]
        - [sp, c1_c2, W_0_0]
        - [sp, c2_pred, W_0_0]
        - [np, conv2, b]
interfaces:
    mnist:
        type: mnist
        in: [mnist_pred]
        out: [mnist_image, mnist_label]
```

```
        remap:
            mnist_image: image
            mnist_label: label
            mnist_pred: prediction
        source_file: /opt/dl/data/mnist.pkl.gz
        fading: 0
tag_specs:
    hidden:
        act: relu
        dropout: 0.2
        device: cuda0
    prediction:
        shape: [10, 1, 1]
        act: softmax
    adam_optimizer:
        device: cuda0
        optimizer: adam
        decay: 0.999
        momentum: 0.99
```

