# OpenReview forum: "Statestream: A toolbox to explore layerwise-parallel deep neural networks"
_ICLR.cc/2018/Conference — Reject_

### Official Review · AnonReviewer2 · 2017-11-27
**The paper describes a toolbox for parallel neuron updating written in Theano.**

**Rating:** 3
**Confidence:** 4

**Review:**

Quality and clarity

The paper goes to some length to explain that update order in a neural network matters in the sense that different update orders give different results. While standard CNN like architectures are fine with the layer parallel updating process typically used in standard tools, for recurrent networks and also for networks with connections that skip layers, different update orders may be more natural, but no GPU-accelerated toolboxes exist that support this. The authors provide such a toolbox, statestream, written Theano.

The paper's structure is reasonably clear, though the text has very poor "flow": the english could use a native speaker straightening out the text. For example, a number of times there are phrases like "previously mentioned", which is ugly.

My main issue is with the significance of the work. There are no results in the paper that demonstrate a case where it is useful to apply fully parallel updates. As such, it is hard to see the value of the contribution, also since the toolbox is written in Theano for which support has been discontinued.

---

> ### Author Response · Authors · 2017-12-22
> **Comment on reviews**
>
> Please see the comment below the first review.

---

### Official Review · AnonReviewer3 · 2017-11-27
**A potentially interesting toolbox not supported by enough examples**

**Rating:** 5
**Confidence:** 4

**Review:**

This paper introduces a new toolbox for deep neural networks learning and evaluation. The central idea is to include time in the processing of all the units in the network. For this, the authors propose a paradigm switch: form layerwise-sequential networks, where at every time frame the network is evaluated by updating each layer – from bottom to top – sequentially; to layerwise-parallel networks, where all the neurons are updated in parallel. The new paradigm implies that the layer update is achieved by using the stored previous state and the corresponding previous state of the previous layer. This has three consequences. First, every layer now use memory, a condition that already applies for RNNs in layerwise-sequential networks. Second, in order to have a consistent output, the information has to flow in the network for a number of time frames equal to the number of layers. In Neuroscience, this concept is known as reaction time. Third, since the network is not synchronized in terms of the information that is processed in a specific time frame, there are discrepancies w.r.t. the layerwise-sequential networks computation: all the techniques used to train deep NNs have to be reconsidered.

Overall, the concept is interesting and timely especially for the rising field of spiking neural networks or for large and distributed architectures. The paper, however, should probably provide more examples and results in terms of architectures that can been implemented with the toolbox in comparison with other toolboxes. The paper presents a single example in which either the accuracy and the training time are not reported. While I understand that the main result of this work is the toolbox itself, more examples and results would improve the clarity and the implications for such paradigm switch. Another concern comes from the choice to use Theano as back-end, since it's known that it is going to be discontinued. Finally I suggest to improve the clarity and description of Figure 2, which is messy and confusing especially if printed in B&W.

---

> ### Author Response · Authors · 2017-12-22
> **Comment on reviews**
>
> Please see the comment below the first review.

---

### Official Review · AnonReviewer1 · 2017-11-27
**Review of "STATESTREAM: A TOOLBOX TO EXPLORE LAYERWISE-PARALLEL DEEP NEURAL NETWORKS"**

**Rating:** 5
**Confidence:** 3

**Review:**

In this paper, the authors present an open-source toolbox to explore layerwise-parallel deep neural networks. They offer an interesting and detailed comparison of the temporal progression of layerwise-parallel and layerwise-sequential networks, and differences that can emerge in the results of these two computation strategies.

While the open-source toolbox introduced in this paper can be an excellent resource for the community interested in exploring these networks, the present submission offers relatively few results actually using these networks in practice. In order to make a more compelling case for these networks, the present submission could include more detailed investigations, perhaps demonstrating that they learn differently or better than other implementations on standard training sets.

---

> ### Author Response · Authors · 2017-12-22
> **Comment on reviews**
>
> We thank the reviewers for their feedback on our work. Considering that responses over reviewers greatly overlapped, we only wrote one comment and put it under the first with a brief note below the other two reviews.
>
> One major concern across reviewers is the lack of compelling examples. We understand and share this concern. Because, we experienced some difficulties in the past explaining the general idea / concept of layerwise parallel networks, we chose to introduce and compare (on a textual level) the two approaches and their implications in some length. On the basis of reviewer's summaries, we think the core idea is well explained (we will try to improve Fig. 1 in the future). Another goal of the paper is to raise awareness inside the community that there are ways to integrate time into networks which are better suited to bridge the gap between spiking and current deep networks than the ones currently used (e.g. rollout or convolution over time).
>
> While we where able to integrate tensorflow support for our toolbox (dependence solely on theano was a concern of two reviewers), we cannot provide meaningful additional examples in the scope of this submission for several reasons: time, pending IP concerns, open technical details, sufficient presentation quality, page restriction.
>
> Again, we want to thank the reviewers for their effort and fair feedback.

---

### Decision · Program_Chairs · 2018-01-29
**ICLR 2018 Conference Acceptance Decision**

**Decision:**

Reject

**Comment:**

This paper presents a toolbox for the exploration of layerwise-parallel deep neural networks. The reviewers were consistent in their analysis of this paper: it provided an interesting class of models which warranted further investigation, and that the toolbox would be useful to those who are interested in exploring further. However, there was a lack of convincing examples, and also some concern that Theano (no longer maintained) was the only supported backend. The authors responded to say that they had subsequently incorporated TensorFlow support, they were not able to provide any more examples due to several reasons: “time, pending IP concerns, open technical details, sufficient presentation quality, page restriction.” I agree with the consensus reached by the reviewers.